# When Two Is Better Than One: A Pilot Study on Transcranial Magnetic Stimulation Plus Muscle Vibration in Treating Chronic Pelvic Pain in Women

**DOI:** 10.3390/brainsci12030396

**Published:** 2022-03-15

**Authors:** Rocco Salvatore Calabrò, Luana Billeri, Bruno Porcari, Loris Pignolo, Antonino Naro

**Affiliations:** 1Neurorehabilitation Unit, IRCCS Centro Neurolesi Bonino Pulejo, Via Palermo-SS113-Cd.a Casazza, 98121 Messina, Italy; bruno.porcari@irccsme.it; 2Istituto Clinico Polispecialistico C.O.T., Cure Ortopediche Traumatologiche s.p.a., Via Ducezio 1, 98100 Messina, Italy; luanabilleri@hotmail.it; 3Brain Injury Center, Sant’Anna Institute, Via Siris 11, 88900 Crotone, Italy; lpignolo@gmail.com; 4AOU Policlinico “G. Martino”—Stroke Unit, Via Consolare Valeria 1, 98125 Messina, Italy; g.naro11@alice.it

**Keywords:** chronic pelvic pain syndrome (CPPS), repetitive transcranial magnetic stimulation (rTMS), supplementary motor area (SMA), pelvic floor muscles (PFMs), focal muscle vibration (FMV)

## Abstract

Chronic pelvic pain syndrome (CPPS) affects about 4–16% of adult women, and about one-third of them require medical assistance due to severe symptoms. Repetitive transcranial magnetic stimulation (rTMS) over the supplementary motor area (SMA) has been shown to manage pain in refractory CPPS. Focal muscle vibration (FMV) has also been reported to relieve pelvic pain. The objective of this study was to assess the feasibility and effect of rTMS coupled with FMV to reduce pain in seven adult women with refractory CPPS. This pilot, open-labeled, prospective trial examined treatment by 5 Hz rTMS over SMA and 150 Hz FMV over the perineum, suprapubic, and sacrococcygeal areas, with one daily session for five consecutive days for three weeks. We assessed tolerance and subjective pain changes (as per visual analog scale, VAS) until one month post-treatment, with a primary endpoint at day 7. No patients experienced serious adverse effects or a significant increase in pain. Six out of seven patients experienced a VAS improvement of at least 10% at T7; three of these individuals experienced a VAS improvement of more than 30%. Overall, we found a significant VAS reduction of 15 points (95% CI 8.4–21.6) at T7 (t = 6.3, *p* = 0.001; ES = 2.3 (1.1–3.9)). Three of the women who demonstrated a significant VAS reduction at T7 retained such VAS improvement at T30. VAS decreased by six points (95% CI 1.3–10.7) at T30 (t = 3.1, *p* = 0.02; ES = 1.5 (0.2–2.6)). This coupled approach seems promising for pain management in adult women with refractory CPPS and paves the way for future randomized controlled trials.

## 1. Introduction

Chronic pelvic pain syndrome (CPPS) is a centralized and disabling pelvic region disorder that is diagnosed when pain has persisted for at least six months. In women, the prevalence of CPPS is between 4 and 16%, and about one-third of them require medical assistance due to severe gastrointestinal and gynecological symptoms, as well as the psychological burden associated with these symptoms [1]. The etiology is multifactorial, including some somatic functional pain syndromes (irritable bowel syndrome, interstitial cystitis, pelvic myofascial pain, bladder pain syndrome, and pudendal neuralgia) and mental health disorders (post-traumatic stress disorder and depression). In most cases, CPPS pathogenesis is related to pelvic floor muscle (PFM) hypertonicity, trigger points in the vulvar area (which are active or latent, small, palpable, hyperirritable nodules located on taut bands of skeletal muscle in areas of sustained contracture), and shortening of the elevator muscles [2,3]. These alterations could lead to a central sensitization disorder, given the continuous triggering of sensorimotor areas.

The management of CCPS is challenging given that pharmacological and rehabilitative treatments (including stretching the muscles of the back, lower limbs, and abdomen; balance and stability training; pelvic re-education; deep massage practices with pressure; joint mobilization; foam rollers; other techniques for releasing the trigger points, such as vibrations, transversal or flat palpation; biofeedback; and transcutaneous electrical stimulation), aimed at promoting muscle relaxation and increasing analgesia, are often unsatisfactory [4]. Moreover, cognitive–behavioral therapy, spinal cord stimulation, total hysterectomy, electrotherapy, short-wave diathermy, respiratory-gated auricular vagal afferent nerve stimulation, percutaneous nerve stimulation, sono-electromagnetic therapy, and ischemic compression at the myofascial trigger point are often therapeutically non-effective [1,3,5,6].

Transcranial magnetic stimulation (TMS) and muscle vibration (MV) are among the complementary approaches to relieve pain [7,8,9]. Repetitive TMS (rTMS) has proven to be effective in various pain types, including nociceptive, neuropathic, and even nociplastic pain [7,10,11,12,13]. Furthermore, it has been suggested that rTMS can reshape the neural mechanism related to pain processing depending on the stimulation site and paradigm [7,8,9]. In particular, it has been shown that rTMS over the supplementary motor area (SMA) may modulate resting PFM activity (tone) [7,14,15,16,17], demonstrating that SMA dysfunction is associated with CPPS by determining an alteration in the cortical silent period duration likely through direct and indirect pathways mediated by periaqueductal gray (PAG), insular, and cingulate cortices [15,18,19,20,21,22,23,24]. However, to date, the effect of rTMS when applied to the PFM representation in SMA has not been systematically tested, and contradicting reports are still available [8].

MV has generated good results concerning muscle strengthening, the normalization of basal muscle tone, and pain relief. However, it has been mainly employed as whole-body vibration [25], whereas the reports on focal MV (FMV) are very sparse [26]. Furthermore, FMV has been mainly employed to treat urinary incontinence and segmental spasticity rather than pelvic pain (but there are reports on exercise-induced pain in athletes and spinal cord-injury-related pelvic pain) [27,28,29,30,31]. In this regard, there is some evidence that the effects of FMV go beyond the local ones, including the modulation of M1 excitability (concerning intracortical and cortical reciprocal inhibition) through both bottom-up (i.e., sensory inputs resetting sensorimotor hyperexcitability) and top-down mechanisms (re-afferent descending volleys from sensorimotor cortex to spinal centers), which, together, favor a reduction in PFM hyper-activation, minimize muscle co-contractions, reduce segmental spasticity and pain, and improve muscle synergies [27,30]. However, even in this case, the usefulness of MV in managing CPPS has not been well established to date, and it has contradicting results.

MV has been combined with other rehab strategies to improve functional outcomes [32]. Combining rTMS and FMV could be useful to augment the individual effects of these approaches. However, to the best of our knowledge, no study has tested the efficacy of this novel protocol in patients with CPPS. This pilot, open-labeled, prospective, clinical study aimed to test this combined approach’s feasibility, efficacy, and safety in women with refractory CPPS, paving the way for a future randomized controlled trial.

## 2. Materials and Methods

### 2.1. Participants and Setting

In this study, we consecutively enrolled female patients attending our rehabilitation institute, who complained of refractory CPPS, between January 2018 and March 2020 (when the enrollment was interrupted due to the COVID-19 pandemic).

The inclusion criteria were the following: (i) A diagnosis of CPPS (i.e., bladder, pudendal, and/or pelvic pain, pressure or discomfort present the majority of the time over the past three months within the previous six months), refractory to the most common treatments, including drugs (antibiotics, nonsteroidal anti-inflammatory drugs, local corticosteroids, antidepressants, and antiepileptic drugs) and surgical treatments; the diagnosis had to be corroborated by a careful clinical examination with urodynamic, laboratory, and urinalysis testing and the exclusion of other syndromes with similar symptoms [33]. (ii) Participants had to be aged between 18 and 65 years.

The exclusion criteria were the following: (i) prior, recent treatment with rTMS; (ii) new medications initiated during the month prior to study entry; (iii) contra-indications to rTMS; (iv) severe pain syndromes other than CPPS; (v) unable to give written informed consent; and (vi) being in the menstrual phase (in such a case, they were deferred).

All study aspects conformed to the principles described in the Declaration of Helsinki and its subsequent amendments, and they were approved by our Institutional Review Board (IRCCSME ID: 32/2017). All participants provided their written informed consent.

### 2.2. Outcome Measures

Patients were assessed daily for one month (i.e., from the thirtieth day prior to treatment onset to the day before treatment onset) (T0) using a visual analog scale (VAS) with a range from 0 to 100, the short form of the Brief Pain Inventory (BPI) [34], the Patient Global Impression of Change (PGIC) [35], Beck’s Depression Inventory (BDI) [36], and Flanagan’s Quality of Life Scale (QOL) [37]. These scales were also administered at the 7-day (T7) and 30-day (T30) follow-up visits.

The primary outcome was the percentage of patients who reported serious adverse effects and who did not complain of a significant decrease in pain (using VAS with a 0–100 range) at T7 compared to T0. A variation of 10% was considered significant [14,17,38,39]. We adopted such a threshold in order to be consistent with previous rTMS studies [14,17,38,39] and according to the spontaneous fluctuation of VAS that we observed during the one-month baseline observation, which was smaller than 10%. Patients were asked to report any adverse effects of stimulation at each session.

Secondary outcome measures included the pain variation (as per VAS) between T30 and T0, and the variations in BPI, PGIC, BDI, and QOL at T7 and T30 compared to T0.

### 2.3. Stimulation Paradigm

Patients underwent an rTMS-MV paradigm every day in the late morning/early afternoon, once a day, five days a week, for three weeks (Figure 1a). Patients were lying supine on a bed, in a mild-lighted room, with their heads on a comfortable pillow, keeping their eyes open.

FMV was delivered to the perineum, suprapubic, and sacrococcygeal areas using the Vibraplus device, a pneumatic vibrator powered by compressed air (@-Circle; San Pietro in Casale, Italy). The device is equipped with cup-like probes of 2 cm^2^ fixed to the suprapubic and sacrococcygeal areas by a Velcro strap and a pen-like probe positioned over the perineum, held by the therapist who carried out FMV (Figure 1b). MV was delivered for 30 min at 150 Hz, with an amplitude (i.e., of the peak-to-peak sinusoidal displacement of the underneath structures) of 4 ± 0.5 mm, which is sufficient to evoke a progressive contraction of the perineal muscles in each of the participants, as assessed by the therapist who carried out the MV.

Once the MV began, we applied a high-frequency rTMS over SMA. In this regard, we first determined the motor hot spot for the relaxed abductor hallucis of the right foot (AH) and first dorsal interosseous of the right hand (FDI) using single magnetic pulses delivered through a figure-of-eight coil (with a 70 mm-diameter loop, connected to a Magstim Bistim2 super-rapid stimulator (the Magstim Co.; Whitland, UK)) placed tangentially over the scalp with the handle pointing backward with a 45-degree angle to the midsagittal for the FDI hotspot and along the sagittal midline with the handle pointing to the right for the AH hotspot (Figure 1c) [40], with a stimulation intensity that elicited the largest motor-evoked potential (MEP) from the target muscle. Then, we determined the active motor threshold (AMT) from FDI and AH by eliciting an MEP of at least 200 μV for 5 out of 10 successive stimuli during voluntary muscle contraction (at 20% of maximum voluntary activity, monitored with EMG) [15]. Then, SMA was identified at 3 cm anteriorly over the midsagittal line to the site of MEP elicitation from AH under active contraction conditions (at 20% of maximum voluntary activity), with a stimulation intensity of 120% AMT (Figure 1c) [41]. Each of the steps mentioned above was repeated before each rTMS session.

Concerning SMA rTMS, we adopted a facilitatory 5 Hz stimulation protocol using the TMS setup mentioned above. Trains of 50 pulses at 5 Hz (10 s) every 50 s, repeated ten times for a total of 500 pulses (in 10 min), were delivered over SMA at 110% AMT of FDI (Figure 1b). We adopted such a protocol as it has been demonstrated to decrease PFM tone due to increased SMA activity [42].

### 2.4. Power Analysis

This pilot study aimed to assess the feasibility and safety of rTMS-MV treatment for refractory CPPS, paving the way for future randomized controlled trials. Therefore, the study sample size was estimated concerning the primary outcome measure (the percentage of patients undergoing the treatment who reported serious adverse effects and did not complain of a significant decrease in pain). A conservative estimate of this effect size was assumed (d = 0.3), given that it widely varies across the published studies [8,43]. Therefore, a sample size of 20 individuals would allow the detection of a lower limit of d > 0.1 for an expected effect size d = 0.3, assuming an α = 0.05 and 1 − β = 0.8 [44]. Five patients should also be included for a 25% dropout.

### 2.5. Statistical Analysis

Continuous data are presented as the mean ± standard deviation or the median (interquartile range) according to statistical distribution (assumption of normality checked using normal probability plots and Shapiro–Wilk’s test).

The assessment of patients tolerating the protocol and showing pain reduction at T7 compared to T0 is presented with a 95% confidence interval and was statistically assessed at the individual level using the reliable change indicator (RCI) [45], which indicates whether or not a reliable change occurred individually.

Paired comparisons were carried out using the Student paired *t*-test or Wilcoxon test. The tests were two-sided, with type I error set at α = 0.05. The Hedges’ g effect size (ES) was calculated [46] and estimated as >0.2—small, >0.5—medium, and >0.8—large effects [47].

The statistical analysis was performed using Stata software, version 13 (StataCorp, College Station, TX, USA).

## 3. Results

Seven women diagnosed with refractory CPPS were enrolled in this pilot study. They all complained of severe pain, mainly localized at the suprapubic region, the pelvis, and the lower abdomen and back. In addition, urinary symptoms, including urgency, frequency, and nocturia, were also reported. VAS individually varied during the one-month baseline assessment by no more than 6 ± 2%, although there were some exacerbation days. However, both these aspects were consistent with the diagnosis of CPPS. Baseline data are summarized in Table 1 and Table 2, and Figure 2. These symptoms negatively affected QOL, despite being actually or formerly appropriately treated with conventional therapy methods.

### 3.1. Primary Outcome Measure

All enrolled patients completed the protocol and were assessed at the one-month follow-up. All patients tolerated the protocol, and none reported any significant side effects or discomfort following each rTMS session, but, occasionally, mild headache and neck pain were reported; in such a case, we slightly increased (+5%) the interval time or slightly reduced (−5%) the duration of stimulation to minimize any side effects. In addition, no pain increase was reported either during or after each rTMS session. Data are summarized in Figure 2 and Table 2.

Six out of seven patients experienced a VAS improvement of at least 10% at T7; three of these individuals experienced a VAS improvement of more than 30% (Table 2). Overall, we found a VAS reduction of about 19 points at T7 (95% CI 8.4–21.6), which was statistically significant at both the group level (t = 6.3, *p* = 0.001; ES = 2.3 (1.1–3.9); Figure 2) and the individual level in six out of seven individuals (RCI values in Table 2).

Thus, the main outcome was achieved with a tolerance rate of 100% (no treatment discontinuation, no significant side effects, and no pain increase) and a significant pain reduction (i.e., more than 10%) in 86% of participants.

### 3.2. Secondary Outcome Measures

Three of the individuals who demonstrated a significant VAS improvement at T7 (i.e., more than 10%) retained such improvement at T30 (43% of participants). Specifically, VAS decreased by about eight points at T30 (95% CI 1.3–10.7), which was statistically significant at both the group level (t = 3.1, *p* = 0.02; ES = 1.5 (0.2–2.6); Figure 2) and the individual level in three out of seven individuals (RCI values in Table 2). Data are summarized in Table 2 and Figure 2.

When assessing pain intensity and interference as per BPI, we found a significant reduction at both T7 (t = 25, *p* < 0.001) and T30 (t = 2.6, *p* = 0.005) (Figure 2). QOL significantly improved at T7 (t = 5.4, *p* = 0.002), whereas such an improvement was not significant at T30 (*p* = 0.2) (Figure 2). BDI significantly improved at T7 (t = 6, *p* = 0.001), whereas such an improvement approached statistical significance at T30 (t = 2.3, *p* = 0.06) (Figure 2). Finally, three patients experienced an improvement, three a partial improvement, and one no improvement according to PGIC at T7. Out of these individuals, three still reported improvement at T30 (Table 2).

## 4. Discussion

For the first time ever, our pilot study suggests that combining rTMS over SMA with FMV to PFM is a safe and effective add-on treatment to alleviate pain and improve QoL in women with refractory CPPS. Pain intensity, as per VAS scoring, was reduced by about 30% in the majority (six out of seven) of patients, with a small effect size (likely due to the small sample) that, however, corresponded to a minimal clinically important difference for chronic pain conditions [48,49,50].

To date, no studies have investigated rTMS over SMA and FMV to PFM applied together. We can, thus, only compare our preliminary findings with the currently available data on stand-alone rTMS over SMA and FMV to PFM. The pain-relieving effects of rTMS applied to PFM representation in SMA are well known, but they require large-scale confirmatory studies [8]. In particular, two rTMS clinical trials on patients with CPPS reported interesting results. A double-blind, sham stimulation-controlled, crossover randomized trial in 13 patients with bladder pain syndrome/interstitial cystitis-related CPPS [17] applied ten 20 Hz rTMS sessions within two weeks (at 110% resting motor threshold, 50 pulses, 30 trains/session, and 30 s of inter-train interval for a total of 1500 pulses/session) with an H-coil over the M1 in the area corresponding to the PFM. The authors achieved a significant VAS decrease compared to the sham stimulation (n = 13) in 3 weeks. In a prospective open-label trial [14], 12 patients with refractory CPPS caused by endometriosis were provided with five 10 Hz rTMS sessions (at 80% resting motor threshold and 50 s of inter-train interval for a total of 1500 pulses/session) targeting the left M1 hand. The authors reported a VAS decrease of 1 point in one month (which was smaller than the minimal clinically important difference). In addition, a case report [18] on high-frequency rTMS carried out in two patients with refractory CPPS, one implanted with a cortical stimulator and the other implanted with a spinal cord stimulator, showed a positive response with a marked pain reduction in two days and partially in 3 weeks after a single rTMS session (the former patient) or two (the latter) rTMS sessions. Finally, a case report [51] on 16 sessions of 1 Hz rTMS sessions (at 110% resting motor threshold for a total of 1200 pulses/session) over both DLPFCs reported a complete resolution of suprapubic pain and a dramatic decrease in micturition frequency.

Overall, these studies suggest that rTMS over SMA may increase or decrease M1 excitability, depending on the SMA conditioning protocol, thus affecting PFM tone [42]. Furthermore, SMA is involved in pain processing neural networks through loops nested within insula and cingulate cortices, which have a significant role in pain processing and neuropathic pain generation [24]. Therefore, it is reasonable that SMA rTMS has a direct or indirect pain-relieving effect. Indeed, the rationale of stimulating SMA to relieve CPPS is supported by the fact that SMA has been reported to be morphologically abnormal and functionally hyperactive in PFM disorders, including CPPS and urinary incontinence [15,18,19,20,21,52,53,54]. Furthermore, PFMs are strongly represented in SMA [24,27]. Finally, SMA interacts with the insular and cingulate cortices, which are relevant in pain control [24]. High-frequency rTMS is proposed to excite SMA and reduce PFM tone; i.e., SMA may exert an inhibitory influence on resting PFM tone [41]. Alternatively, SMA may be reset by a high amount of rTMS-induced facilitation, thus inhibiting PMF tone. rTMS may reshape intracortical inhibitory circuits in such a way, likely via subcortical dopamine–opioid neural networks [55].

FMV is a validated approach to manage pain and spasticity-related pain in different neurological conditions, including stroke, multiple sclerosis, and cerebral palsy [30,32,56,57]. CPPS has been mainly treated using whole-body MV [58], whereas, to date, FMV has not been applied to PFM, except in the context of erectile dysfunction and PFM spasticity in young men with chronic, incomplete, post-traumatic spinal cord injury [30]. Additionally, FMV has been adopted to manage drooling in children with cerebral palsy [59], post-stroke shoulder spasticity [32], and diabetic neuropathic pain [60]. In the aforementioned pilot study [30], 15 sessions of 30 min FMV to PFM, suprapubic, and sacrococcygeal areas (delivered three times weekly for five consecutive weeks) provided the patients with a PMF spasticity decrease and a subjective pain improvement lasting about three months, which were paralleled by an amplitude increase in the electrophysiological bulbocavernosus reflex and the pudendal nerve somatosensory-evoked potentials. Pain reduction could be indeed directly due to the spasticity improvement. However, the sensory stimulation provided by FMV may gate afferent nociceptive inputs from the pelvic floor at both the spinal and cortical levels. Additionally, it could interfere with neuropathic pain generation in CPPS [60]. The FMV-induced sensory input repeatedly reaches M1 via Ia fibers, entraining intrinsic plasticity-related mechanisms that, per se, lead to an improvement in motor function [32,56,61,62]. In particular, FMV may imply both a non-synaptic (involving changes in the intrinsic properties of neural membranes, likely affecting the motor threshold) and a synaptic Hebbian-like plasticity mechanism (which may account for neural pathway cross-activation and cortical maps) [63]. In addition, FMV locally increases nitric oxide production and improves angiogenesis and blood flow, which are all mechanisms involved in pain relief [64,65,66]. Despite all these mechanisms that may account for the long-lasting changes in pain suffering induced by FMV to PFM, the rationale of its application requires further validation analyses since it is entirely new concerning pain relief in CPPS.

The aftereffects of the 15 rTMS and FMV sessions that we employed were longer than those reported in the abovementioned rTMS studies [14,17,18,51], were significant at the one-week follow-up visit in the majority of patients, and were still appreciable up to the one-month follow-up visit in about half of the patients. On the contrary, stand-alone FMV effects have been reported to last up to three months [30]. We thus hypothesize that combining FMV with rTMS could have favored both the magnitude and the duration of the aftereffects. One may argue that repeating rTMS stimulation cycles could have merely elongated the stimulation aftereffects [7] without the necessity to complement rTMS with FMV. However, shorter rTMS protocols have been associated with a partial, positive response three weeks after treatment [17]. Therefore, providing more rTMS sessions does not necessarily guarantee more lasting aftereffects than the combined approach. The interaction between the two protocols sustaining the more lasting and greater effects as compared to the stand-alone approaches (although a direct effect size comparison cannot be reliably performed due to clear differences in samples and study protocols) may occur at both the cortical level, where the proprioceptive inputs (preferentially through Ia afferents) offered by FMV may affect the intracortical inhibitory systems (i.e., a bottom-up mechanism through which sensory inputs reset sensorimotor hyperexcitability), thus affecting corticospinal excitability and muscle synergy (i.e., a top-down mechanism through which re-afferent descending volleys from the sensorimotor cortex affect the spinal centers), and at the spinal level, where the proprioceptive inputs result in the inhibition of the monosynaptic reflex and the restoration of abnormal reciprocal and presynaptic inhibition mechanisms [30,32,56,57]. Independently, the timing and duration of the aftereffects of the combined approach suggest that a synaptic strengthening was achieved over a short period compared to the stand-alone rTMS paradigms reported in the literature [10,67,68,69], owing to a reciprocal potentiation of both stimulation techniques. Our results are nonetheless interesting if we consider that the patients enrolled in the study experienced pain for a long time, our short, combined protocol being sufficient to integrate the conventional therapy for refractory CCPS compared to longer rTMS or FMV stand-alone protocols.

Finally, we have to acknowledge the potential interaction among neuromodulation and concomitant pharmacological therapy. Even though we did not assess or observe (although we may expect it in larger samples) a decrease in “drug consumption for pain” or “number of days taking drugs for pain”, an interaction between the treatments is plausible. In fact, there could be a synergic effect between neuromodulation and conventional therapy, including cyclo-oxygenase inhibitors, tricyclic antidepressants, and neuromodulation drugs (e.g., gabapentin and pregabalin), which can all affect pain-processing-related synaptic plasticity [70,71,72,73]. According to the putative mechanisms exerted by rTMS and FMV at the central nervous system level, we may hypothesize that neuromodulation may help pharmacological therapy, by a sort of Hebbian-like associative plasticity, to drive the pain-processing-related neuronal and glial plasticity mechanisms that should be triggered by the drugs themselves [43,74], but they do not occur due to patient-related factors, among others [75]. Furthermore, appropriate treatment is nevertheless mandatory, as it can be assumed that psychological support and the appropriate treatment of anxious–depressive symptoms may both reduce the number of possible relapses and increase rTMS + FMV efficacy in the long term [76].

The main study limitations are the lack of a control group and the limited sample size. However, this study was intended as a pilot one to preliminarily assess the safety and feasibility of the approach. In addition, the importance of a placebo effect deserves further investigation, even though there were no evident BDI changes, thus suggesting that the clinical alleviation of pain was not preponderantly due to improvements in the patients’ psychological statuses [39].

## 5. Conclusions

The management of refractory CPPS is still challenging despite the availability of several pharmacological and non-pharmacological approaches, thus raising the need for new treatment options. Our pilot study suggests that coupling high-frequency rTMS over SMA with FMV to PMF is feasible, effective, and well tolerated in patients with refractory CPPS. A consistent proportion of the patients reported a significant improvement in individually estimated pain and PGIC up to 30 days following the end of the stimulation protocol. The coupled approach, together with actual pharmacological therapy, may positively influence pain by mediating a functional reorganization of the pain-related plasticity processes related to the emotional component of pain, the pain processing mechanisms, and the descending inhibitory pathways. Furthermore, this combined approach potentially fits with other therapies, as it has no pharmacological or surgical side effects and does not interfere with concomitant pharmacological treatments. Consistent with the limited sample and the pilot nature of the study, our data pave the way for future, larger, randomized clinical trials to support this approach as a new analgesic technique to be applied in daily clinical practice. This will also allow more accurate and thorough knowledge of the neurophysiological correlates of rTMS and FMV concerning pain reduction in CPPS.

## Figures and Tables

**Figure 1 brainsci-12-00396-f001:**
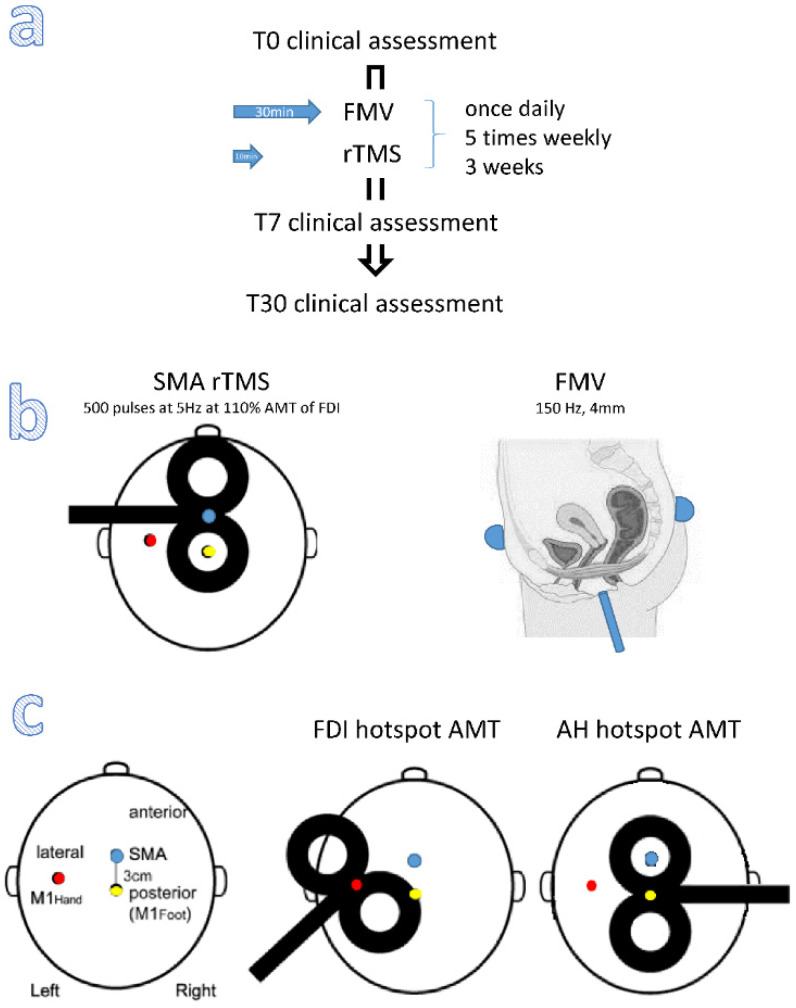
(**a**) Experimental study design. (**b**) SMA rTMS site and cup-like and pen-like probe sites for FMV. (**c**) Single-pulse TMS stimulation sites. Legend: AH: abductor halluces; AMT: active motor threshold; FDI: first dorsal interosseous; FMV: focal muscle vibration; M1: primary motor cortex; rTMS: repetitive transcranial magnetic stimulation; SMA: supplementary motor area; T0: baseline; T30: follow-up visit 30 days after the end of the stimulation protocol; T7: follow-up visit seven days after the end of the stimulation protocol.

**Figure 2 brainsci-12-00396-f002:**
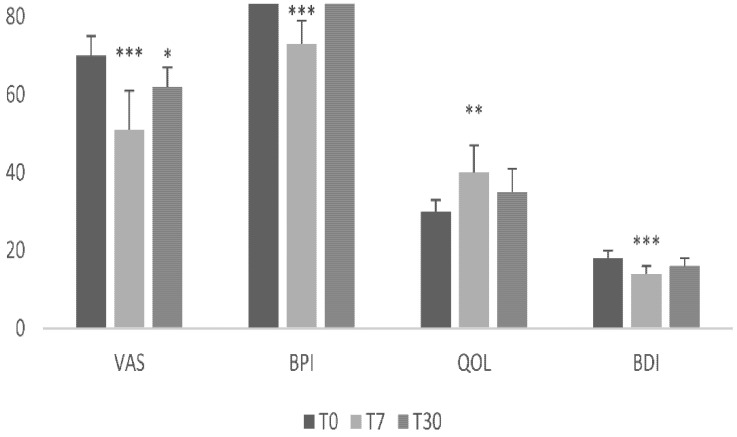
Average values of the outcome measures. Data are reported as mean with standard deviation; * indicates a significant T7-T0 or T30-T0 change (*** *p* < 0.001; ** *p* < 0.01; * *p* < 0.05). Legend: VAS: visual analog scale; T0: mean scores of the 30 days before the first treatment session; T7: 7-day follow-up visit. T30: 30-day follow-up visit; BPI: Brief Pain Inventory short form; BDI: Beck’s Depression Inventory; QOL: Flanagan’s Quality of Life.

**Table 1 brainsci-12-00396-t001:** Patients’ characteristics. Data are reported as mean with standard deviation or count. Legend: NSAID: non-steroidal anti-inflammatory drug.

Age (Years)	Disease Duration (Years)	Pain Treatment
35	8	no
42	5	tramadol
56	13	NSAID pregabalin
37	8	no
55	12	amitriptyline
32	11	NSAID pregabalin
34	9	no
42 ± 10	9 ± 3	4 treated 3 not treated

**Table 2 brainsci-12-00396-t002:** Individual visual analog scale (VAS) and Patient Global Impression of Change (PGIC) values. Data are reported as mean with standard deviation, percentage change, or count. VAS includes raw data and the percent reduction at T7 and T30 compared to T0 (with the relative RCI value; * whether significant, i.e., >1.96). Legend: VAS: visual analog scale; T0: mean scores of the 30 days before the first treatment session; PGIC: patient global impression of change; T7: day 7 following the end of the stimulation protocol; T30: day 30 following the end of the stimulation protocol; RCI: reliable change index.

T0	T7				T30			
				PGIC				PGIC
68	39	−42%	6.4 *	much improved	64	−6%	0.9	no change
73	53	−27%	4.4 *	minimally improved	70	−4%	0.6	no change
60	39	−35%	4.7 *	improved	56	−7%	0.9	no change
70	49	−30%	4.7 *	improved	59	−17%	2.6 *	minimally improved
75	59	−22%	3.8 *	minimally improved	62	−17%	2.9 *	minimally improved
72	65	−9%	1.5	no change	68	−5%	0.9	no change
70	55	−22%	3.5 *	minimally improved	59	−17%	2.6 *	minimally improved
70 ± 5	51 ± 10	−27 ± 11%		3 improved 3 partially improved 1 no improvement	62 ± 5	−10 ± 6%		3 partially improved 4 no improvement



## Data Availability

The data presented in this study are available on request from the corresponding author.

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
