# Peer review of "When Two Is Better Than One: A Pilot Study on Transcranial Magnetic Stimulation Plus Muscle Vibration in Treating Chronic Pelvic Pain in Women"

_brainsci, 2022, doi:10.3390/brainsci12030396_

Round 1

Reviewer 1 Report

The work by Calabrò and colleagues investigated the feasibility and effect of coupled repetitive transcranial magnetic stimulation (rTMS) and focal muscle vibration (FMV) to reduce pain in women with refractory chronic pelvic pain syndrome (CPPS).

Overall, this is a good study. The writing, presentation and experimental approach are scientifically sound. Discussion and introduction section are very well written. There are however some points to consider which I think will improve the understanding and coherence of this manuscript.

I suggest that table 1 be reorganized. At this point it is very difficult to read the data.

Please correct the typos; there are a lot of them in the whole manuscript.

All patients tolerated the protocol and none reported any significant side effect or discomfort (but, occasionally, mild headache and neck pain) following each rTMS session. No pain increase was reported either during or after each rTMS session. In this context, how were the side effects minimized?

How do you think conventional therapy for refractory CCPS may impact coupled rTMS and FMV?

The conclusions are very general and I would expect them to be more to the point.

Author Response

The work by Calabrò and colleagues investigated the feasibility and effect of coupled repetitive transcranial magnetic stimulation (rTMS) and focal muscle vibration (FMV) to reduce pain in women with refractory chronic pelvic pain syndrome (CPPS). Overall, this is a good study. The writing, presentation and experimental approach are scientifically sound. Discussion and introduction section are very well written. There are however some points to consider which I think will improve the understanding and coherence of this manuscript.

We thank the reviewer for the appreciation of our ms and the helpful suggestion to improve its quality.

  • I suggest that table 1 be reorganized. At this point it is very difficult to read the data.

Accordingly, Table 1 was entirely rebuilt and integrated with Figure 2 and Table 2.

  • Please correct the typos; there are a lot of them in the whole manuscript.

The ms was amended consistently.

  • All patients tolerated the protocol and none reported any significant side effect or discomfort (but, occasionally, mild headache and neck pain) following each rTMS session. No pain increase was reported either during or after each rTMS session. In this context, how were the side effects minimized?

We added the missing information. We slightly increased the interval time (+5%) or slightly reduced (-5%) the duration of stimulation to minimize any side effect.

  • How do you think conventional therapy for refractory CCPS may impact coupled rTMS and FMV?

We thank the reviewer for this food for though. We did not assess or observe, but we may expect it in larger samples, a decrease in “drug consumption for pain” and “number of days taking drugs for pain”. We also added a comment on a possible synergistic effect between neuromodulation and conventional therapy, including cyclo-oxygenase inhibitors, tricyclic antidepressants and novel neuromodulation drugs (gabapentin, pregabalin), which can all affect pain processing-related synaptic plasticity. Furthermore, an appropriate pharmacological treatment is anyway mandatory, as it can be assumed that psychological support and appropriate treatment of anxio-depressive symptoms may both reduce the number of possible relapses and increase rTMS+FMV efficacy in the long term.

  • The conclusions are very general and I would expect them to be more to the point.

We better expanded the conclusion section, as suggested.

Kindest regards,

the authors

Reviewer 2 Report

Lines 25-26 need to be rephrased. Perhaps something like “Out of the individuals who demonstrated a significant VAS reduction at T7, three retained such VAS improvement at T30.”

Lines 56- Select another word rather than “useless” perhaps therapeutically noneffective, or clarify term "useless" in terms of alleviating CCPS-related pain.

Lines 209-210  Clarify “pain reduction rate of 85% as the percent changes presented in Table 1 were all much lower than 86%.  Where is this percentage coming from? Only 3/7 had changes in pain over 30%?

Lines232-233  Clarify “pain reduction rate of 43% as the percent changes presented in Table 1 were all much lower than 43%.  Only 3/7 had changes in pain over 30%.

Line 120. Assessment were taken 30 days prior to treatment onset, but was an assessment taken again just before treatment began? Is T0 the assessment at 30 days prior to onset of treatment or was it the day before or day of onset treatment.  Why wasn't an assessment repeated just before the onset of treatment? If a 10% change is considered “significant” could there not have been a 10% fluctuation during the 30 days prior to treatment onset?  It is a rather small threshold to be considered a significant change in pain. If two baseline assessments were not taken (one 30 days prior to onset and a 2nd baseline just before treatment began) this would allow you to assess normal fluctuation. With lack of a control group, how do you know the amount of  natural fluctuation over time does not reach 10%.  This should be considered as a limitation, in  addition to the lack of a control group as mentioned.

Table 1 should be spread out more, there appears to be enough space, and it would make it much more easier to read.

Line329-330  Rephrase “further interesting”

Author Response

We thank the reviewer for the helpful suggestion to improve the quality of our ms.

  • Lines 25-26 need to be rephrased. Perhaps something like “Out of the individuals who demonstrated a significant VAS reduction at T7, three retained such VAS improvement at T30.”

Corrected as suggested.

  • Lines 56- Select another word rather than “useless” perhaps therapeutically noneffective, or clarify term "useless" in terms of alleviating CCPS-related pain.

Corrected as suggested.

  • Lines 209-210 Clarify “pain reduction rate of 85% as the percent changes presented in Table 1 were all much lower than 86%.  Where is this percentage coming from? Only 3/7 had changes in pain over 30%?

The sentence was revised since misleading; we specified “a significant pain reduction (i.e., more than 10%) in 86% of participants”.

  • Lines232-233 Clarify “pain reduction rate of 43% as the percent changes presented in Table 1 were all much lower than 43%.  Only 3/7 had changes in pain over 30%.

The sentence was revised since misleading; we specified “Out of the individuals who demonstrated a significant pain improvement at T7, three retained such improvement at T30 (a significant pain reduction, i.e., more than 10%, in 43% of participants)”.

  • Line 120. Assessment were taken 30 days prior to treatment onset, but was an assessment taken again just before treatment began? Is T0 the assessment at 30 days prior to onset of treatment or was it the day before or day of onset treatment. Why wasn't an assessment repeated just before the onset of treatment? If a 10% change is considered “significant” could there not have been a 10% fluctuation during the 30 days prior to treatment onset?  It is a rather small threshold to be considered a significant change in pain. If two baseline assessments were not taken (one 30 days prior to onset and a 2nd baseline just before treatment began) this would allow you to assess normal fluctuation. With lack of a control group, how do you know the amount of  natural fluctuation over time does not reach 10%.  This should be considered as a limitation, in  addition to the lack of a control group as mentioned.

We thank the reviewer for giving us the opportunity to better clarify this point, which was described in the paper in a misleading way. We now better specified that patients were daily assessed for one month for each of the outcome measures in order to evaluate, as correctly pointed out by the reviewer, the normal fluctuation of the outcome measures. This data was added to the ms in the result section limitedly to VAS, which was the primary outcome measure. We understand the reviewer’s concern that a 10% change is a rather small threshold to be considered a significant change in pain. Furthermore, the minimal clinically important difference has been reported to be 15-30%, depending although on pain etiology and onset (acute vs. chronic). We adopted such a threshold in order to be consistent with previous rTMS works and according to the spontaneous fluctuation of VAS during the one-month baseline observation we observed, which was rather smaller than 10% (as now better highlighted in the text).

  • Table 1 should be spread out more, there appears to be enough space, and it would make it much more easier to read.

Accordingly, Table 1 was entirely rebuilt and integrated with Figure 2 and Table 2.

  • Line329-330 Rephrase “further interesting”

Corrected.

Kindest regards,

the authors.